# Molecular Insights into the Marine Gastropod *Olivancillaria urceus*: Transcriptomic and Proteopeptidomic Approaches Reveal Polypeptides with Putative Therapeutic Potential

**DOI:** 10.3390/ijms26083751

**Published:** 2025-04-16

**Authors:** Gabriel Marques de Barros, Letícia Fontes Gama, Felipe Ricardo de Mello, Claudia Neves Corrêa, Louise Oliveira Fiametti, Horácio Montenegro, Cristiane Angélica Ottoni, Leandro Mantovani de Castro

**Affiliations:** 1Department of Biological and Environmental Sciences, Bioscience Institute, São Paulo State University (UNESP), Sao Vicente 11330-900, SP, Brazil; gabriel.m.barros@unesp.br (G.M.d.B.); leticia.fontes@unesp.br (L.F.G.); felipe.r.mello@unesp.br (F.R.d.M.); claudia.neves@unesp.br (C.N.C.); cristiane.ottoni@unesp.br (C.A.O.); 2Biodiversity of Coastal Environments Postgraduate Program, Department of Biological and Environmental Sciences, Bioscience Institute, São Paulo State University (UNESP), Sao Vicente 11330-900, SP, Brazil; 3Institute of Chemistry, São Paulo State University (UNESP), Araraquara 14800-060, SP, Brazil; louise.fiametti@unesp.br; 4School of Pharmaceutical Sciences, São Paulo State University (UNESP), Araraquara 14800-903, SP, Brazil; 5NGS Soluções Genômicas, Piracicaba 13416-030, SP, Brazil; h.montenegro@gmail.com

**Keywords:** omics, marine gastropod, bioprospection

## Abstract

The marine environment is a rich source of new biotechnologies and products. Bottom trawling for shrimp species such as *Xiphopenaeus kroyeri* and *Farfantepenaeus brasiliensis* leads to the unintentional capture of non-target species, known as bycatch, which includes a variety of marine life that are often discarded without economic value. A common bycatch species on the southeast coast of Brazil is *Olivancillaria urceus* (*O. urceus*), a carnivorous gastropod that feeds mainly on bivalves. Despite its abundance, this species is still little studied, especially for biotechnological applications. Other gastropods such as Conus are known for their diverse and potent toxins, which offer great potential for pharmacological discoveries. In this study, an omics approach, including transcriptomics and proteopeptidomics, was applied to explore *O. urceus* at the molecular level. The transcriptome of the muscle foot/mantle led to the annotation of 19,097 genes via Gene Ontology, identifying 20 toxin-like transcripts identified considering the Gastropod class. The proteome fraction confirmed 2179 transcripts, including sequences with toxin activity, such as conotoxin precursors, Conodipine-P3, and BPTI/Kunitz domain-containing proteins. In addition, 9663 peptides of 1484 precursor proteins were detected in the peptide fraction, including 2 sequences representing neurotoxins. The identification of these sequences could lead to the discovery of new molecules with therapeutic potential.

## 1. Introduction

Bottom trawling for the *Xiphopenaeus kroyeri*, *Farfantepenaeus brasiliensis*, and *Farfantepenaeus paulensis* shrimp species is a low-selectivity extraction method that results in the unintentional capture of non-target species, known as bycatch, including fish, turtles, crustaceans, mollusks, sponges, corals, and others [1,2]. These incidental catches are considered fishing byproducts that often have no recognized economic value and are discarded lifeless [3,4,5,6].

Among the species frequently captured as bycatch in bottom trawling on the south-east coast of Brazil is *Olivancillaria urceus* (*O. urceus*), a mollusk from the family Olividae and the class Gastropoda, which was first described by Röding in 1798. Endemic to the Southeast Atlantic, this species has been documented as an abundant bycatch [1]. Its distribution range extends from the Bahia state in Brazil to the Gulf of San Matias in Argentina and inhabits shallow sandy bottoms in neritic waters at depths ranging from 0 to 53 m [7]. These organisms are carnivorous and feed mainly on bivalves [8]. They live in symbiosis with other microorganisms, such as Gram-positive bacteria, including Firmicutes and Actinobacteria, and Gram-negative bacteria, including Proteobacteria [9]. Regarding their morphology, they have a dark pink foot and siphon, and their shell has a subquadrangular shape; dirty pink coloration with axial and spiral lines; and an irregular brown, ochre-colored fasciolar band that can be up to 63.5 mm long, which is thick, smooth, and polished, with a maximum thickness of 1 mm (Figure 1A,B) [7]. Currently, there are not many studies searching for molecules from this neglected resource, and no in-depth molecular characterization has been performed.

Recently, whole-body peptide extracts of the congeneric species *O. hiatula* showed antimicrobial activity against both Gram-positive and Gram-negative bacteria, with minimum inhibitory concentrations (MICs) ranging from 0.039 to 2.5 mg/mL [10]. Furthermore, another study by the same research group observed antibiofilm activity with 50% inhibition against *Pseudomonas aeruginosa*, starting at 39 µg/mL of the peptide extract [11].

In the last twenty years, advances in high-throughput sequencing techniques for nucleic acids (DNA/RNA) and proteins/peptides have enabled an increase in the knowledge of non-model species [12,13,14,15]. Most of these recent studies have taken an integrated approach incorporating transcriptomics and utilizing RNA-seq to create databases representing the coding regions of the genome [16,17,18], along with proteomics and peptidomics data obtained via high-performance liquid chromatography coupled with mass spectrometry [19].

Using these molecular tools, natural marine products (NMPs) have been characterized as bioactive molecules, particularly for therapeutic use. The classes of NMPs include peptides, which exhibit a broad spectrum of biological activities, including cytotoxic antineoplastic [20], antimicrobial [21], analgesic [22], and antiviral activities [23], among others.

An example of research on natural marine products (NMPs) in mollusks is the conotoxin, a diverse group of neurotoxic peptides isolated from the venom of marine gastropods of the genus Conus. Conotoxins are known for their exceptional specificity and potency in targeting ion channels, receptors, and transporters in the nervous system. These peptides are classified into subfamilies based on their molecular targets, including voltage-gated sodium (NaV), calcium (CaV), and potassium (KV) channels, as well as nicotinic acetylcholine receptors (nAChRs) [24,25]. Structurally, conotoxins are small peptides that are typically 8 to 30 amino acids in length, and they are stabilized by multiple disulfide bridges that form specific cysteine frameworks (e.g., CC-C-C, C-C-CC, C-C-C-C), which are essential for their structural rigidity and precise interactions with molecular targets. Conotoxins are synthesized as inactive precursors and undergo post-translational modifications, such as proteolytic cleavage, disulfide bridge formation, and C-terminal amidation, to become biologically active. Their mechanism of action involves selective binding to ion channels or receptors, modulating their activity and disrupting neurotransmission. For instance, α-conotoxins block nAChRs, leading to paralysis; μ-conotoxins inhibit NaV channels, preventing the propagation of action potentials; ω-conotoxins target CaV channels, blocking the release of neurotransmitters; κ-conotoxins inhibit KV channels, affecting neuronal excitability [26].

In addition to their use in research, conotoxins have immense potential for therapeutic applications. One of the most well-known conotoxins, Ziconotide (Prialt^®^), derived from *Conus magus*, has been approved for clinical use as a powerful non-opioid analgesic. Ziconotide blocks N-type voltage-gated calcium channels and thus effectively relieves chronic pain by inhibiting the release of neurotransmitters in the pain pathway [27,28,29].

Considering the problem of bycatch fauna and the potential for molecule bioprospecting in the marine environment, this study aimed to add information about a discarded marine gastropod through a range of integrated omics techniques.

## 2. Results

### 2.1. Transcriptome

Illumina sequencing generated a total of 239,009,224 paired reads (2 × 119,504,612) of 100 bases each. Transcriptome assembly by Trinity resulted in 319,067 transcripts, and protein prediction by TransDecoder yielded 238,999 predicted peptides (108,636 complete proteins, 75,458 5′ partial peptides, 28,826 3′ partial peptides, and 26,079 internal peptides). We analyzed the BUSCO results based on the integrity of our transcriptome and compared it to universally conserved orthologs. The analysis was performed in two stages: one using Trinity to assemble the transcriptome and the other using TransDecoder to predict the proteins. BUSCO analyses revealed a high level of completeness for both the transcriptome and the predicted proteome, with 93.0% of the 954 expected orthologs identified in each dataset. Specifically, Trinity recovered 524 (54.9%) single-copy orthologs, while TransDecoder identified 540 (56.6%), suggesting a good level of assembly integrity and accurate protein prediction. However, a relatively high proportion of duplicated complete orthologs was observed—363 (38.1%) in Trinity and 347 (36.4%) in TransDecoder—potentially reflecting biological gene duplications or assembly artifacts. Fragmented orthologs were comparatively few, with 43 (4.5%) in Trinity and 47 (4.9%) in TransDecoder. Additionally, only 24 (2.5%) and 20 (2.1%) orthologs were missing in the Trinity and TransDecoder datasets, respectively.

Based on the protein prediction by Trinotate, the total number of transcripts with any prediction exceeds the number of transcripts identified by Trinity, because a single transcript sequence can generate more than one possible protein isoform. Furthermore, a large number of transcripts were observed to have no annotation. Out of the 378,795 identified transcripts, 301,180 did not match any sequences in the databases included in this analysis (approximately 75.51% of the total transcriptome; see Appendix A). The NR_Diamondx_BLASTX database had the highest number of hits, covering just over 18% of the transcriptome (Figure 2).

From these data, a filtering process was applied to obtain unique genes related to Gene Ontology (GO). The result was 19,097 genes characterized by GO and summarized by WEGO. Among these, 16,357 were associated with cellular components, 16,182 with biological processes, and 16,869 with molecular functions, resulting in a total of 49,408 GO terms. The subcategorization of the annotation based on the GO terms provided by WEGO showed a consistent distribution with respect to cellular components, with the majority originating from the terms “Cell” with 15,112 genes and “Organelle” with 12,405 genes (Figure 3A). In terms of molecular function, the highest concentration of sequences was related to “binding” with 13,717 genes and “catalytic activity” with 7908. Additionally, sequences associated with “toxin activity” were observed, with a total of 46 genes providing information of interest for bioprospecting (Figure 3B). To assess the expression of these toxins across all transcripts, we performed a transcripts per million (TPM) analysis. This allowed us to assess the presence of toxin-related transcripts within the Gastropoda class. The distribution of these transcripts is shown in Table 1, where the highest TPM values were observed for transcripts DN27199 c2 g2 i1 (371,809), DN16518 c0 g1 i1 (36,110), and DN3301 c0 g1 i1 i2 (10,069). Finally, subcategorization into biological processes showed that most sequences are involved in cellular processes (14,532); 11,061 are involved with metabolic processes and 9465 are involved with biological regulation (Figure 3C).

In addition, an analysis was conducted to assess the phylogenetic proximity of proteins predicted by TransDecoder, aiming to characterize genera closely related to the genus Olivancillaria. A primary filter was applied to the Gastropoda class, followed by separation relative to the genus (Figure 4).

### 2.2. Proteopeptidome

Proteomic analysis was performed on foot muscle/mantle protein extracts of *O. urceus*, with a single LC/MSMS run for each sample (n = 4) of the fraction above 10 kDa. A total of 2179 proteins were identified, considering an FDR < 5%, using the database generated by Transdecoder based on protein prediction from the transcriptome (Appendix A). Of the total proteins identified via mass spectrometry, only 18.54% (404 proteins) did not have any known annotations and can be classified as unknown or uncharacterized. A total of 678 proteins detected via mass spectrometry were present in all groups. The demonstration of the overlapping relationships between different groups can be seen in Figure 5. The statistical results of protein identification are shown in Appendix A.

Regarding the databases, most proteins were identified through entries in the nr_diamond_BLASTX database and nr_diamond_BLASTP, accounting for approximately 89.1% of the total proteins (Figure 6).

Using Gene Ontology (databases BLASTP, BLASTX, and Pfam), the proteins could be classified into three main categories: cellular component, biological process, and molecular function. For cellular components, 1601 Gene Ontology terms were obtained; for biological processes, 1497 terms were found; for molecular functions, 1659 terms were identified. For a total of 1067 proteins, there were no corresponding Gene Ontology terms (Appendix A). Next, Figure 7 shows the five subcategories with the highest number of Gene Ontology occurrences for each previously established category. Finally, proteomic analyses enabled the detection of three sequences related to toxin activity: DN16518_c0_g1_i1, DN27199_c2_g2_i1, and DN3301_c0_g1_i1_i2. These toxins present in the proteome showed high expression in the transcriptome in terms of transcripts per million compared to other toxins (Table 1; Figure 8).

For peptidomics analysis, an LC-MS/MS run was conducted for each of the four samples of the fraction below 10 kDa, representing naturally occurring peptides. A total of 9663 peptides from 1484 precursor proteins were found, considering an FDR of <5% and using the database generated by Transdecoder from the transcriptome. Of the total number of precursor proteins identified via mass spectrometry, only 14.65% (314 precursor proteins) had no annotation and could be classified as unknown or uncharacterized. As for the databases, most of the precursor proteins were identified by entries in the nr_DIAMOND_BLASTX database, corresponding to about 88% of the total proteins.

Similarly to the proteomic analysis, we used Gene Ontology (gene_ontology_BLASTP, gene_ontology_BLASTX, and gene_ontology_Pfam databases) for the peptidome and categorized the peptides into three main groups, cellular component, biological process, and molecular function, followed by subcategories for each of these factors. For cellular components, we obtained 1061 Gene Ontology terms; for biological processes, 1015 terms were found; for molecular function, 1089 terms were found. For a total of 754 peptides, there were no Gene Ontology terms associated with them. Appendix A shows the distribution of subcategories with respect to the highest number of occurrences according to the main categories; this is considered with more than 30 occurrences for better graphic visualization.

To deepen the analysis of the subcategories, we selected the five subcategories with the highest number of Gene Ontology occurrences for each previously identified category and used the Pfam database to observe which families and/or protein domains these subcategories are associated with. To better visualize the data, for the cellular component and molecular function categories, we only considered categories with an occurrence of more than 20, while for the biological process, an adjustment to an occurrence of more than four was necessary as the data were relatively sparse compared to the previous categories (Figure 9).

In addition, we carried out a more comprehensive analysis by searching beyond the SWISS-Prot/UniProt database to identify potential toxins in the peptidomics data. As a result, we found two transcripts with Pfam matches: DN31687_c0_g1_i1 matched Toxin_9 (PF02819.17), and DN29606_c0_g1_i2 matched Toxin_12 (PF07740.14; Figure 10).

## 3. Discussion

This study is the first to describe a dataset of transcripts, proteins, and peptides for the marine gastropod species *O. urceus*, specifically focusing on the foot muscle/mantle region, using different omics methods. Regarding general transcriptome data, a high number of transcripts that did not have any type of annotation was observed (approximately 75.51% of the total transcriptome). This peculiarity in the data was expected, as there is no molecular database for this nor for a few neighboring species and genera. This reaffirms the absence of molecular data from non-model organisms, so there was a need to evaluate the data via phylogenetic proximity, mainly considering the Gastropoda class.

The transcriptome, comprising 319,067 transcripts from Trinity and 235,860 polypeptides predicted with the TransDecoder tool, provided a crucial dataset for analysis. A similar number of transcripts has also been observed in other gastropod species [30,31,32]. The experimental validation of these predictions was supported by proteomic and peptidomic data obtained through mass spectrometry, which revealed 2179 proteins—1656 unique to the proteome and 961 unique to the peptidome. It is worth highlighting that 523 proteins were common across all three datasets (Figure 11), underscoring their robust expression and consistency across different analytical approaches.

Primarily, we identified an abundance of proteins and enzymes mainly related to biological processes such as actin cytoskeleton organization, cell proliferation, cell adhesion, sarcomere components, and signal transduction. The protein classes found include filamins, spectrins, ankyrins, cadherins, actins, myosins, tropomyosins, and tubulins, as well as domains related to calcium ion binding, such as calmodulin, troponin C, and EF-Hand. In addition, the presence of a high amount of collagen was also detected. The high expression of these proteins is justified by the type of tissue analyzed, predominantly composed of the muscular foot and mantle regions, which consist mainly of muscle and connective tissue [33,34]. However, these tissues in gastropods are also covered by an epidermal layer with different types of epithelial cells, ranging from cuboidal to columnar shapes, which can be ciliated or mucus-secreting, and they are associated with functions such as locomotion, predation, and protection [35].

Additionally, our analyses also allowed the identification of proteins and peptides with putative sequences related to toxin activity. The availability of molecular data on toxins from the genus Conus played an important role in the study of potential toxins present in the transcriptome of *O. urceus*. Of the transcripts related to toxin activity, 18 showed similarities with the genus Conus and are discussed below (Table 2).

### 3.1. Putative Toxin-Related Transcripts and Polypeptides in O. urceus

Transcript DN16518 c0 g1 i1 shows high expressions and sequence similarities to Conodipine-P1, while transcripts DN19184 c2 g1 i1 and DN19184 c2 g1 i2 exhibit similarities to Conodipine-P1. Additionally, transcript DN13158 c3 g1 i1 shows similarities to Conodipine-P3. These bioactive proteins, originally isolated from the venom of *Conus purpurascens*, are members of the phospholipase A2 family, specifically the Group IX subfamily. Structurally, both Conodipine-P1 and P3 are heterodimers consisting of alpha and beta chains connected by disulfide bridges, which are crucial for protein stability and functionality. The disulfide bridges also confer resistance to proteolytic degradation, enhancing the functional integrity of the toxins in diverse environments [36] (Table 2).

The DN27199 c2 g2 i1 transcript correlates with a conotoxin precursor, sharing structural motifs typical of conotoxins, including a hydrophobic signal domain, a propeptide, and a cysteine-rich mature peptide. Post-translational modifications, such as cleavage into three domains and disulfide bridge formation, are critical for its maturation and biological activity [37,38] (Table 2).

The transcript DN3301 c0 g1 i1 exhibited a sequence similar to Kunitz-type domains found in various biological systems, such as animals, plants, and fungi; these are primarily described as serine protease inhibitors and toxins in venomous animals [39,40,41]. Structurally, this type of domain consists of 50 to 70 amino acid residues, and they are organized into two antiparallel β-sheets and stabilized by three disulfide bridges with a connection pattern of C1–C6, C2–C4, and C3–C5 [42] (Table 2). As observed in the amino acid sequence, the transcript detected in *O. urceus* contains two of these domains and the region F*Y*GC****N*F*****C, which is a signature sequence of the Kunitz family [43]. Additionally, BLAST (https://blast.ncbi.nlm.nih.gov/Blast.cgi, accessed on 30 November 2024) analysis showed 55.47%, 49.26%, and 47.79% identity with the TFPI-like multiple Kunitz-type protease inhibitor sequence from the salivary gland of the non-conoidean neogastropod *Colubraria reticulata* (SPP68599.1) [44] and with conkunitizins from *Conus magus* (DAC80551.1) and *Conus ermineus* (AXL95648.1), respectively [42,43].

The DN109957 c0 g1 i1, DN126505 c0 g1 i1, DN133748 c0 g1 i1, and DN17279 c3 g1 i1 transcripts are similar to Conotoxin Im14.3 from *Conus imperialis*, which is composed of 73 amino acids and stabilized by two disulfide bridges. The cysteine framework XIV (C-C-C-C) conducts the arrangement of disulfide bonds, which are essential for maintaining the three-dimensional conformation of the neurotoxin [45] (Table 2).

ConoCAP-related transcripts DN132248 c0 g1 i1 and DN4345 c0 g2 i2 encode a 207 amino acid protein that is processed into three different chains: ConoCAP-a, ConoCAP-b, and ConoCAP-c; moreover, post-translational modifications occur, such as the formation of disulfide bridges and amidation, which are important for the protein’s functionality [46] (Table 2).

The DN18169 c0 g1 i3 and DN18169 c0 g1 i2 transcripts are associated with Elevenin-Vc1, a toxin consisting of 100 amino acids. Structurally, this toxin belongs to the Elevenin family and forms a monomer stabilized by disulfide bridges. Its C-C cysteine scaffold is essential for its neuropeptide-like function, affecting the nervous system of prey [47] (Table 2).

Tereporin-Ca1, identified from transcripts DN55000 c0 g1 i1 and DN87107 c0 g1 i1, is a 190-amino acid pore-forming protein. This toxin, which belongs to the actinoporin family, forms 1 nm wide pores in lipid membranes, a structural feature associated with cytolytic and cardiac stimulatory functions [48] (Table 2).

The DN60625 c0 g1 i1 transcript correlates with Perivitellin-2, a heterodimeric protein consisting of a 31 kDa tachylectin subunit and a 67 kDa MACPF subunit. Structurally, the two subunits form a pore-forming complex with an internal diameter of approximately 5.6 nm, which contributes to its cytotoxic effect in mammalian cells [49] (Table 2).

Turripeptide Pal9.2, which is associated with the DN46921 c0 g1 i1 transcript, comprises 70 amino acids and contains three disulfide bridges. This neurotoxin targets ion channels and shares structural features with Kazal-type serine protease inhibitors, which confer additional protease resistance [50] (Table 2).

The DN31687 c0 g1 i1 transcript was identified in the peptidome fraction with a 60.4% identity similarity to the protein LOC112566798 (XP_025098946) from *Pomacea canaliculata* and obtained a TPM of 3706. This uncharacterized protein was predicted via automated computational analysis and is part of the genomic annotation of BioProject PRJNA472795 [51]. The 98 amino acid sequence shows a remarkable region between residues 73 and 92 that has similarities to spider neurotoxins such as agatoxin, purotoxin, and ctenitoxin, suggesting a possible neurotoxic function [52].

The DN29606 c0 g1 i2 transcript was identified in the Pfam database. It is associated with the PF07740 domain, has a TPM of 457, and is described as an inhibitory ion-channel toxin. This family of potent toxins acts as ion-channel inhibitors, blocking different types of ion channels [53].

Although these transcripts show sequence similarities to known toxins and contain conserved structural motifs associated with bioactive proteins, their functional roles in *O. urceus* have not yet been demonstrated. Future studies aiming to experimentally validate the activity, expression patterns, and potential ecological functions of these sequences will be essential to clarify their biological relevance in this species.

**Table 2 ijms-26-03751-t002:** Summary of the putative toxin-related transcripts and polypeptides from *O. urceus* identified by omics compared to toxins described in Conus and their respective actions.

Transcript ID—*O. urceus*	Name of the Toxin in Conus	Probable Toxic Activity	Author
DN16518 c0 g1 i1	Conodipine-P1	Phospholipase A2	[36]
DN19184 c2 g1 i1			
DN19184 c2 g1 i2			
DN 13158 c3 g1 i1	Conodipine-P3		
DN27199 c2 g2 i1	Conotoxin precursor Pmag02	Conotoxin	[37,38]
DN3301 c0 g1 i1	Kunitz-type domain	Serine protease inhibitors/toxin	[42,43]
DN109957 c0 g1 i1	Conotoxin Im14.3	Likely as a neurotoxin	[45]
DN126505 c0 g1 i1			
DN133748 c0 g1 i1			
DN17279 c3 g1 i1			
DN132248 c0 g1 i1	ConoCAP	Decreases heart rate	[46]
DN4345 c0 g2 i2			
DN18169 c0 g1 i3	Elevenin-Vc1	Toxin induces hyperactivity	[47]
DN18169 c0 g1 i2			
DN55000 c0 g1 i1	Tereporin-Ca1	Function of a pore-forming protein	[48]
DN87107 c0 g1 i1			
DN60625 c0 g1 i1	Perivitellin-2 protein (31 kDa subunit)	Cytotoxicity	[49]
DN46921 c0 g1 i1	Turripeptide Pal9.2 toxin	Inhibiting ion channels by similarity with other similar toxins	[50]
DN12249 c1 g1 i1	Thyrostimulin alpha-2 subunit	Function of toxin by similarity	[54,55]
DN94413 c0 g1 i1	Thyrostimulin beta-5 subunit		

### 3.2. Limitations of Omics Analyses

Although we identified 20 toxin transcripts in the Gastropoda class from our transcriptome database, most of these transcripts represented the potential diversity of toxins expressed only at the mRNA level. When we correlated these transcripts with the data obtained via proteomics and peptidomics, we found some significant differences. Only transcripts DN27199 c2 g2 i1, DN16518 c0 g1 i1, DN3301 c0 g1 i1, DN31687 c0 g1 i1, and DN29606 were identified in the proteomics and peptidomics analyses (Figure 8 and Figure 10), which can be attributed to biological and technical factors.

Firstly, mRNA expression does not always translate directly into protein production. Post-transcriptional regulation, such as mRNA degradation, translation efficiency, and protein stability, plays critical roles in determining the final proteins that are present in the cell. In addition, some proteins can be expressed at very low levels, making them difficult to detect [56,57].

In addition, the sensitivity and coverage of proteomics technologies can limit the detection of low-abundance or highly hydrophobic proteins, such as some toxins. Proteomic and peptidomic experiments confirmed the expression of many genes, particularly those with high transcript abundance per million. However, some less abundant transcripts were not detected and identified in these analyses, which can be in part explained by certain technical factors with respect to the mass spectrometry method, such as the ionization capacity of specific peptide sequences, the overlap of distinct peptides at the same elution time, and the priority for detecting ions in higher abundances [58,59,60]. Finally, the differential expression of toxins may be specific to certain physiological states, environmental conditions, and stages of the life cycle, and sampling may not have captured these specific moments [61,62].

## 4. Materials and Methods

### 4.1. Collection and Maintenance of Specimens

The specimens of *O. urceus* were collected during shrimp trawling carried out with bottom trawls in Peruíbe County on the south coast of São Paulo State (−24.346556, −46.956072; Figure 1C). The specimens were placed in a container with seawater, sand, and an aerator and taken to the laboratory. The acclimatization period lasted 7 days, with a photoperiod of 12 h, a salinity of 33 ppm in artificial seawater, and 25 °C temperature. The specimens were fed two or three times a week with small fragments of squid.

### 4.2. RNA Extraction, mRNA Library Synthesis, and Illumina Sequencing

For transcriptomic analysis, a specimen of *O. urceus* was placed on ice, and the muscular foot/mantle region was removed with scissors. After tissue removal, the sample was immediately stored in liquid nitrogen for shipping to NGS Soluções Genômicas, Piracicaba, Brazil. Due to the specific aim of using the transcriptome as a reference database for proteomic analysis, only a single specimen was used, allowing us to prioritize a higher sequencing depth. After the addition of Rnase-free Dnase I to eliminate genomic DNA, RNA extraction was performed following the Rneasy Lipid Tissue Kit protocol. RNA quality was assessed using an Agilent Bioanalyzer (Santa Clara, CA, USA). The sample had an absorbance ratio of 260/280 nm > 1.8 and an RIN value of 7.8, indicating sufficient RNA integrity for subsequent analyses. The cDNA library was synthesized with ~1 μg of total RNA, following the recommendations of the RNA TrueSeq manufacturer. The resultant cDNA libraries were sequenced using an Illumina HiSeq 2500 platform. The Real-Time Analysis program (Illumina, San Diego, CA, USA) was used to perform the base calling of sequencing images, converting them into fastq sequences. Before assembly, raw reads were evaluated using FastQC [63].

### 4.3. Transcriptome Assembly, Annotation and Functional Enrichment

The resultant sequencing reads were submitted to de novo assembly using Trinity version 2.11.0, a pipeline for transcriptome assembly. Standard assembly parameters were used with a minimum length of 50 bp. Trinity filtered bases of low quality, removed Illumina adapters with Trimmomatic version 0.36 [64], counted k-mers present in sequencing reads with Jellyfish version 2.3.0 [65], and performed digital normalization to reduce data volumes without affecting the assembly’s accuracy. Next, the modules Inchworm, Chrysalis, and Butterfly were used to assemble the transcripts; reconstruct possible transcript isoforms; and group them into “genes”. Trinity was executed with default options plus the argument “--trimmomatic --SS_lib_type RF”. The annotation of the transcriptome was performed using Trinotate pipeline version 3.2.2, which integrates various tools to predict proteins from transcriptome assembly to obtain functional annotation across multiple gene categories. Initially, TransDecoder version 5.5.0 [17] was used to extract the longest open reading frames (ORFs) from the transcripts and translate these ORFs in silico into their respective peptides. The peptides obtained in the previous step were used in similarity searches performed with DIAMOND version 2.0.11 [66] against the SwissProt/UniProt protein database [67], HMMER version 3.2.2 [68], and the Pfam database [69]. Peptides longer than 50 amino acids and peptides showing similarities to the databases were retained for further analyses. The retained transcripts were analyzed for the prediction of the start and stop codons and possible splicing sites, resulting in the final prediction of peptides. The transcripts assembled using the Trinity methodology and the peptides predicted by TransDecoder were compared in similarity searches performed with DIAMOND against the SwissProt and NCBI NR databases [70].

The assessment of the assembled transcriptome quality was obtained using BUSCO program version 5.1.3 [71], which evaluates genome quality via the presence of an expected set of conserved single-copy genes in the assembly. Evaluation gene sets were derived from the OrthoDB database version 10.0 [72], representing a list of single-copy orthologous genes found in all representatives of certain taxonomic groups. The ortholog database used for analysis pertained to the Metazoa kingdom. The assembled transcriptome by Trinity (with the parameter “--mode trans”) and the predicted proteome by TransDecoder (with the parameter “--mode prot”) were also analyzed.

Functional annotations were performed by comparing the sequences with those in public databases. Gene Ontology (GO) terms were assigned to each unigene based on the GO terms annotated to their corresponding homologs. Unigenes were classified according to GO terms within molecular functions, cellular components, and biological process categories, and they were additionally plotted using WEGO 2.0 (Web Gene Ontology Annotation Plot) software with default parameters.

### 4.4. Proteomics and Peptidomics Procedures

#### 4.4.1. Protein and Peptide Fraction Preparation

For proteomics and peptidomics, a total of 12 specimens were used, and they were divided into four groups based on shell size. Groups 1, 2, and 3 had similar body sizes, and in group 4, the animals were smaller. Shell lengths ranged from 3.5 to 5.0 cm, while the maximum width varied from 2.2 to 3.2 cm (Appendix A).

The protein and peptide fractions of foot muscle/mantle from *O. urceus* were prepared as previously described, with some modifications [13]. Collected tissues were resuspended in 10 mL of deionized water at 80 °C each, and they were homogenized with a mechanical tissue homogeneizer (Thomas Scientific, Logan, NJ, USA) and kept at the same temperature for 20 min. After incubation, the samples were cooled at 4 °C on ice for 40 min and then acidified with HCl to a final concentration of 10 mM. Cooled samples were sonicated 10 times with 20 pulses of 1 s at 4 Hz, and the homogenates were centrifuged at 5000× *g* for 40 min at 4 °C. After this point, supernatants were filtered through a Millipore membrane that allows the passage of molecules with a molecular weight of less than 10 Kda (Amicon Ultra, Millipore, Burlington, MA, USA). Eluates not retained by the membrane (peptide fraction) were loaded onto C18 pre-columns (OASIS—Waters, UK), washed with deionized water, and eluted within 100% acetonitrile containing 0.15% trifluoroacetic acid (TFA). Sample volumes were concentrated to 5 μL in a vacuum centrifuge and stored at −80 °C. Samples retained by the membrane (protein fraction) were processed for proteomic analysis. A solution of 8 M Urea was added to the incubation sample to a final concentration of 4 M, followed by the addition of dithiothreitol (DTT) to a final concentration of 5 mM. The mixture was incubated at 65 °C for 60 min. Iodoacetamide (IAA) was then added to a final concentration of 15 mM, and the samples were incubated for 60 min at room temperature in the dark. To quench the excess of IAA, DTT was added to a final concentration of 10 mM; proteins were digested with respect to 1:50 trypsin–sample (Promega, Madison, WI, USA) overnight at 37 °C, acidified with formic acid, and desalted.

#### 4.4.2. NanoLC and Mass Spectrometry

Peptide mixtures were suspended in 0.1% formic acid and analyzed as follows. An UltiMate 3000 Basic Automated System (Thermo Fisher^®^, Waltham, MA, USA) was set up and connected online with a Fusion Lumos Orbitrap mass spectrometer (Thermo Fisher^®^, Waltham, MA, USA) at the mass spectrometry facility RPT02H/Carlos Chagas Institute—Fiocruz, Paraná. Peptide mixtures were chromatographically separated on a column (15 cm in length with an internal diameter of 75 μm) and packed in-house with ReproSil-Pur C18-AQ 3 μm resin (Dr. Maisch GmbH HPLC, Ammerbuch, Germany) at a flow rate of 250 nL/min of 5% to 38% ACN in 0.1% formic acid on a 120 min gradient. Fusion Lumos Orbitrap was placed in the data-dependent acquisition (DDA) mode to automatically turn between full-scan MS and MS/MS acquisition with 60 s dynamic exclusion. Survey scans (300–1500 *m*/*z*) were acquired in the Orbitrap system with a resolution of 120,000 at *m*/*z* 200. The most intense ions captured in a 2 s cycle were chosen, excluding those that were unassigned or had a 1+ charge state. The selected ions were then isolated in a sequence and fragmented using HCD (higher-energy collisional dissociation) with a normalized collision energy of 30%. The fragment ions were analyzed with a resolution of 50,000 at 200 *m*/*z*. The general mass spectrometric conditions were as follows: 2.3 kV spray voltage, no sheath or auxiliary gas flow, heated capillary temperature of 175 °C, predictive automatic gain control (AGC) enabled, and an S-lens RF level of 30%. Mass spectrometer scan functions and nLC solvent gradients were regulated using the Xcalibur 4.1 data system (Thermo Fisher^®^, Waltham, MA, USA).

#### 4.4.3. Protein and Peptide Identification

Raw data files (.raw) generated via the mass spectrometer were searched in the Transdecoder database built from *O. urceus* using the PEAKS Studio software (version 8.5; Bioinformatics Solution, Waterloo, ON, Canada) [73,74]. The research parameters used were as follows: no enzymatic specificity relative to peptidome, trypsin, and proteome fractions; precursor mass tolerance adjusted to ±15 ppm; fragmentation ion mass with a tolerance of ±0.5 Da. Oxidized methionine (+15.99 Da) and acetylation (+42.01 Da) were defined as variable modifications. Carbamidomethylation (+57.02 Da) was also added as a variable modification for alkylated and reduced samples. The identified peptides were then sorted by their mean local confidence to select the best spectra for annotation and filtered by FDR ≤ 5%.

#### 4.4.4. Integrative Data Analyses of the Transcriptome and Proteome

The database generated by TransDecoder, containing amino acid sequences predicted from transcriptome data and identified by the Trinity code, was used as a reference for identifying MS and MS/MS fragments. This process enabled the determination of peptide sequences, validating the transcriptome data. With the confirmation of TRINITY/TRANSDECODER identification codes in proteopeptidomics, the integration was carried out as described below.

To analyze and integrate the transcriptome and proteome data, R programming language was used in the RStudio environment, version 4.4.0. In addition to the base package, the following were used: “ggplot2” version 3.5.1 [75], “dplyr” version 1.1.4 [76] (https://github.com/tidyverse/dplyr, accessed on 13 March 2024, https://dplyr.tidyverse.org), “tidyverse” version 2.0.0, “readxl” version 1.4.3 [75], and “stringr” version 1.5.1 [77]. R package version 1.5.1 [78] (https://github.com/tidyverse/stringr, accessed on 13 March 2024; https://stringr.tidyverse.org) packages were used. Initially, the data were imported using the readxl package, which makes it easier to read Excel files, allowing large volumes of transcriptome and proteome data to be imported. Next, the dplyr package was used to efficiently manipulate these data, enabling operations such as data selection, filtering, and aggregation. The stringr package was used to manipulate strings, which is essential for processing and standardizing transcript and protein identifiers. Data integration was carried out using tidyverse, a set of packages that offers tools for data science, allowing data to be organized and analyzed in a structured way. To visualize the results, the ggplot2 package was used, which allows graphs to be created and is used to visually represent the analyses, making it easier to interpret data.

## 5. Conclusions

In summary, the omics of *O. urceus* reflect a comprehensive set of molecular adaptations that enable its survival in a dynamic and challenging marine environment. These analyses revealed a protein profile dominated by several key families essential to cellular processes, such as cytoskeleton organization, cell signaling, and adhesion, underscoring the critical roles of these functions in adaptation. Furthermore, this study confirmed the detection of transcripts and polypeptides related to toxin activity, highlighting the importance of using integrative approaches to study gene expression in a non-model organism. Together, these findings emphasize not only the biological functions of *O. urceus* but also the biotechnological potential of its proteins for future research and applications.

## Figures and Tables

**Figure 1 ijms-26-03751-f001:**
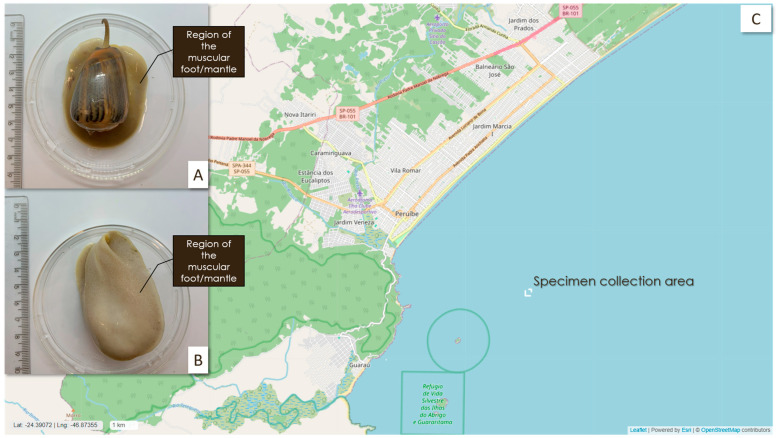
The images show (**A**) the shell of the species and (**B**) the underside of the animal, highlighting the anatomical region from which the fragments were collected for transcriptomic and proteomic analysis. The map (**C**) shows the geographical region where the *O. urceus* samples were collected; Peruíbe County on the south coast of São Paulo State (−24.46556, −46.956072).

**Figure 2 ijms-26-03751-f002:**
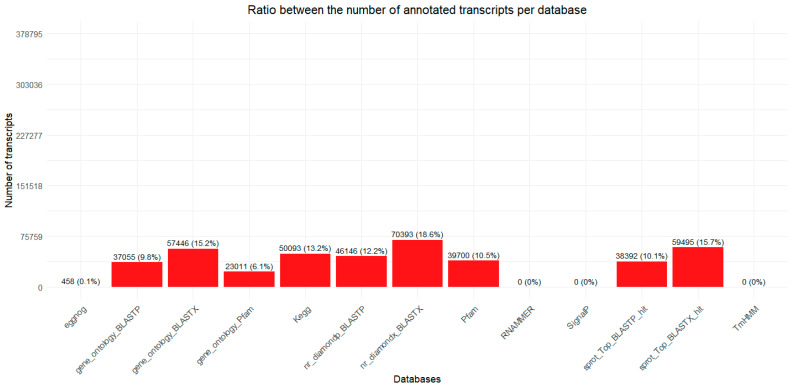
Percentage of identified and annotated transcripts of the *O. urceus* per database. The databases included in the study were as follows: eggNOG, gene_ontology_BLASTP, gene_ontology_BLASTX, Gene_Ontology_Pfam, KEGG, nr_diamond_BLASTP, nr_diamond_BLASTX, PFAM, RNAMMER, SignalP, sprot_Top_BLASTP_hit, sprot_Top_BLASTX_hit, and TrmHMM.

**Figure 3 ijms-26-03751-f003:**
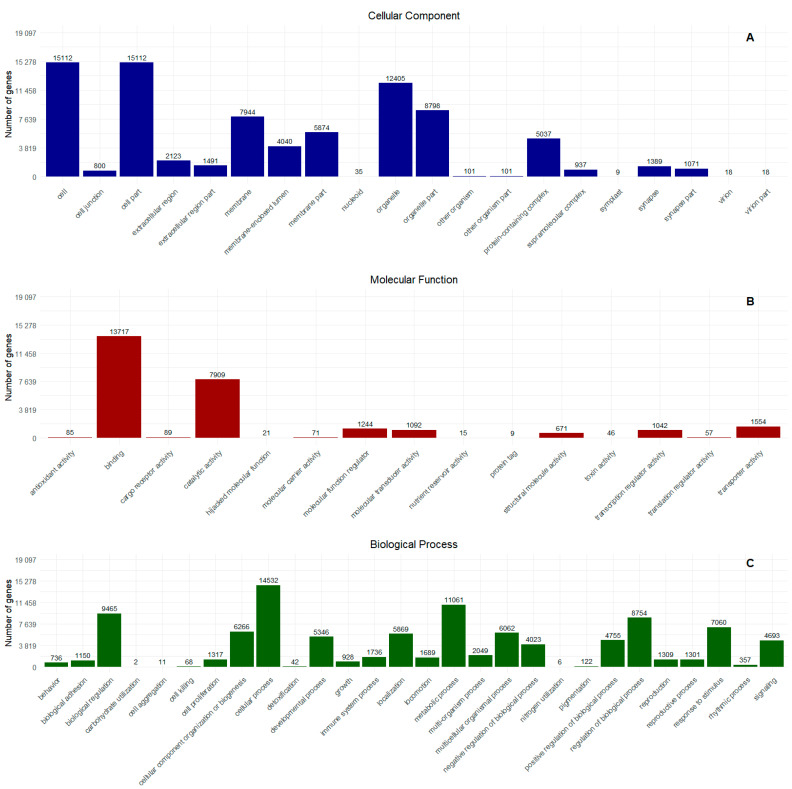
Analysis of the functional categories based on the Gene Ontology of transcripts obtained from *O. urceus*. (**A**) Cellular components, (**B**) molecular functions, and (**C**) biological processes.

**Figure 4 ijms-26-03751-f004:**
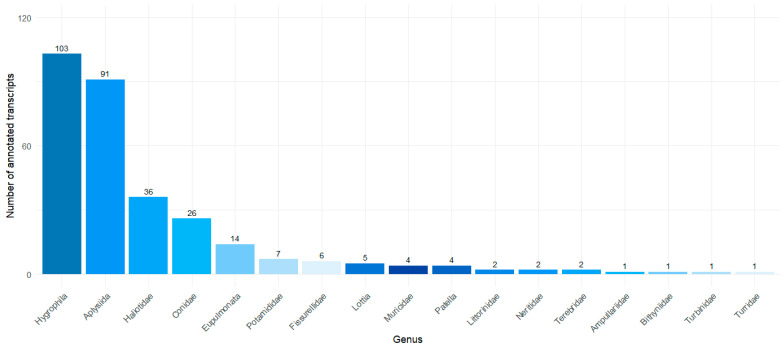
Number of annotated transcripts per genus in the Sprot_BLASTP database.

**Figure 5 ijms-26-03751-f005:**
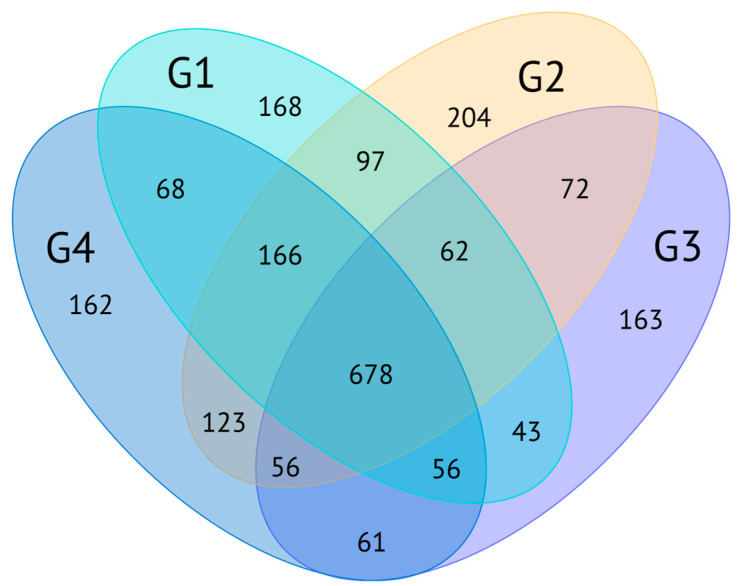
Number of identified proteins and overlap between sample groups in the proteomic analysis. Each group (G1, G2, G3, and G4) corresponds to a sample processed with a pool from the foot/mantle region of three animals. The details of the animals in each group are shown in Appendix A.

**Figure 6 ijms-26-03751-f006:**
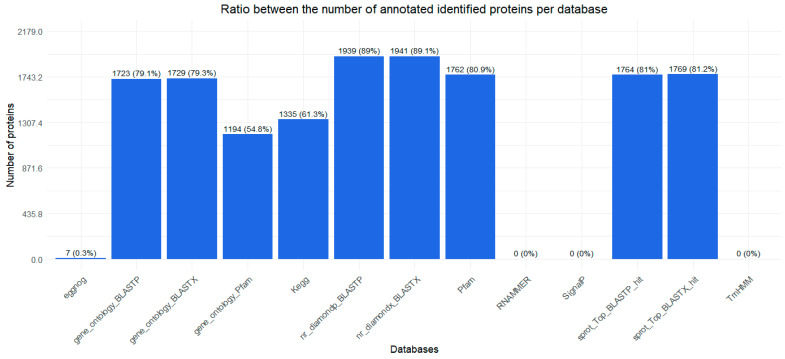
Percentage of identified and annotated proteins per database. The bar chart shows the number of proteins identified (Y-axis) in different databases (X-axis). The percentage within each bar represents the annotated proteins in relation to the total number of analyzed proteins. The databases included in the study are eggNOG, gene_ontology_BLASTP, gene_ontology_BLASTX, Gene_Ontology_Pfam, KEGG, nr_diamond_BLASTP, nr_diamond_BLASTX, PFAM, RNAMMER, SignalP, sprot_Top_BLASTP_hit, sprot_Top_BLASTX_hit, and TrmHMM. This graph shows the coverage of each database in the annotation of proteins identified from the *O. urceus* transcriptome.

**Figure 7 ijms-26-03751-f007:**
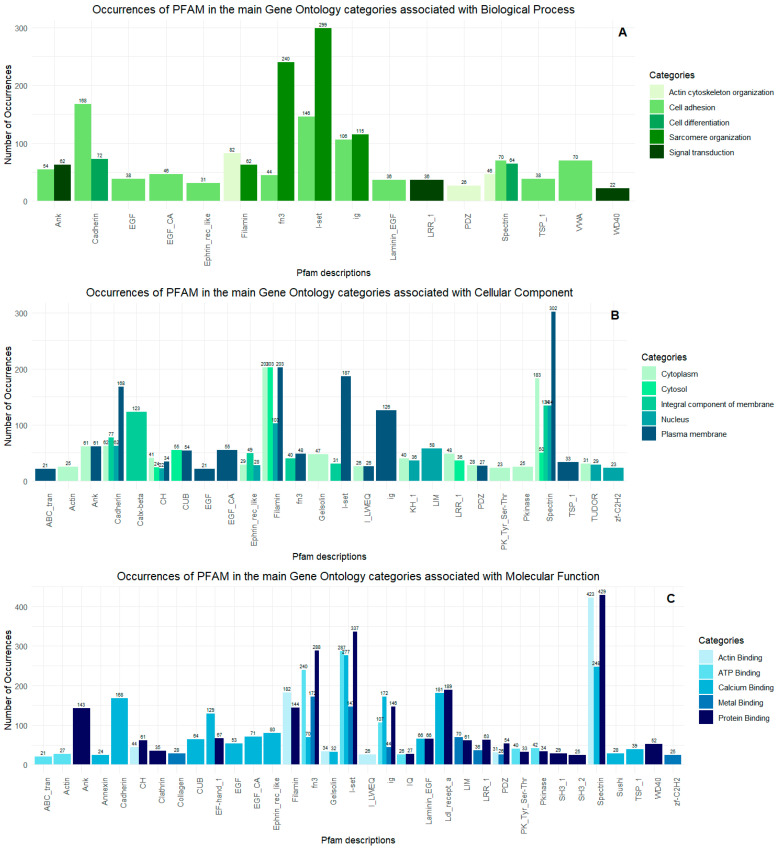
Occurrences of PFAM domains in the main Gene Ontology categories: (**A**) biological process, (**B**) cellular components, and (**C**) molecular functions observed in the proteome. In (**A**), the graph shows the number of occurrences in key biological processes, such as actin cytoskeleton organization, cell adhesion, cell differentiation, and signal transduction. In (**B**), the domains are presented in categories related to cellular components, including the cytoplasm, cytosol, integral membrane components, nucleus, and plasma membrane. Finally, (**C**) highlights domains involved in molecular functions, such as actin binding, ATP binding, calcium ion binding, metal ion binding, and protein binding.

**Figure 8 ijms-26-03751-f008:**
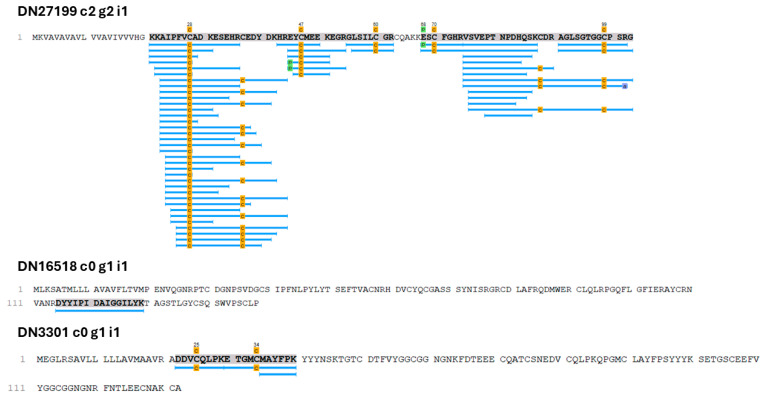
Peptide fragments identified as toxin-like proteins in the proteomic analysis of *O. urceus*. The top sequence, DN27199 c2 g2 i1, shows similarities to DAZ86947.1, identified as “Conotoxin precursor Pmag02”. Highlighted peptide coverage includes *KKAIPFVCAD…AGLSGTGGCPSRG*. The middle sequence, DN16518 c0 g1 i1, resembles COP3_CONPU, classified as “Conodipine-P3”. Highlighted coverage includes the *DYIIPIDAIGGILYK* region. The bottom sequence, DN3301 c0 g1 i1, corresponds to KCP_HALAI from the “BPTI/Kunitz domain” family. Highlighted peptide coverage includes *DDVCQLPKETGMCMAYFPK*. Modification Sites: Letter c, highlighted in dark yellow: Carbamidomethyl (57.02 Da) on Cysteine; Letter p, highlighted in green: Pyro-glu from E (−18.01 Da), N-terminal; Letter a, highlighted in light blue: Amidation (−0.98 Da), C-terminal. Amino acid residues in bold with a blue underline indicate peptides identified by proteomics. Numbers (e.g., 1, 111) indicate the position of the first amino acid of each line within the full protein sequence.

**Figure 9 ijms-26-03751-f009:**
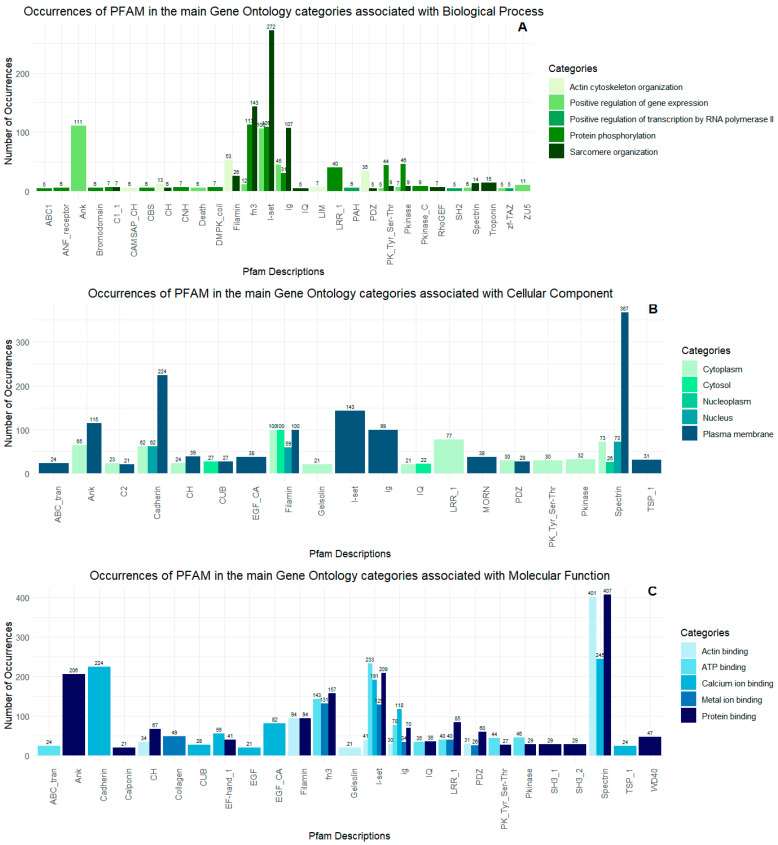
Occurrences of PFAM domains in the main Gene Ontology (GO) categories identified in the *O. urceus* peptidome. (**A**) Biological process, highlighting the predominance of proteins involved in actin cytoskeleton organization, the regulation of transcription by RNA polymerase II, and protein phosphorylation. (**B**) Cellular component. (**C**) Molecular function, emphasizing proteins associated with actin, ATP, calcium ions, and metals.

**Figure 10 ijms-26-03751-f010:**
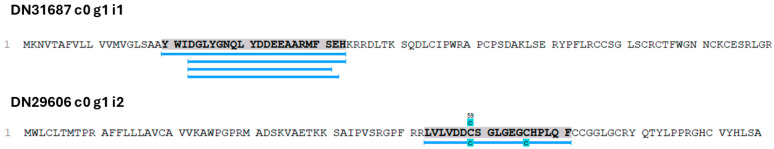
Peptide sequences identified via the peptidomics analysis of *O. urceus*. The top sequence, DN31687 c0 g1 i1 p1, shares 60.4% identity with XP_025098946, an uncharacterized protein from *Pomacea canaliculata*. This sequence was also classified in Pfam as Toxin_9 (PF02819.17). The highlighted region indicates the following peptide sequence coverage: *YWIDGLYGNQLYDDEEAARMFSEH*. The bottom sequence, DN29606 c0 g1 i2 p2, corresponds to PF07740, part of the “Toxin_12” family. The peptide sequence coverage is highlighted as *LVLVDDCSGLGGECCHPLQF*, with detected post-translational modifications (carbamidomethylation at positions C59 and C66; letter c, highlighted in blue). Amino acid residues in bold with a blue underline indicate peptides identified by proteomics. Number 1 indicate the position of the first amino acid of each line within the full protein sequence.

**Figure 11 ijms-26-03751-f011:**
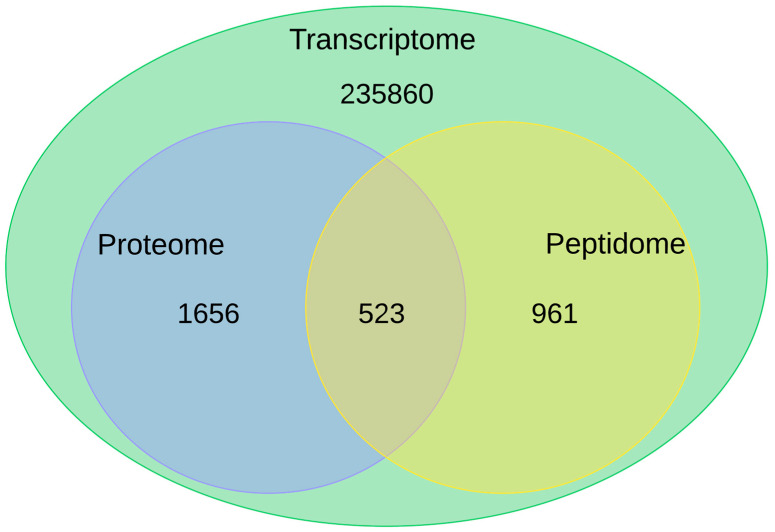
Integrated omics from foot muscle/mantle of *O. urceus*. Comparative analysis of transcripts confirmed by proteomics and peptidomics techniques.

**Table 1 ijms-26-03751-t001:** Transcripts identified in the *O. urceus* transcriptome and annotated genes from UniProt/Swiss-Prot BLASTP related to toxin activity by the GO term in the Gastropoda class.

Transcript ID (Trinity ID)	Uniprot/Swiss-Prot ID	Alignment Region in the Sequence	Similarity	*p*-Value	Description	TPM *
DN27199 c2 g2 i1	DAZ86947.1	Q:6-98, H:3-86	33.1%	1 × 10^−6^	Conotoxin precursor Pmag02	371,809
DN16518 c0 g1 i1	COP3_CONPU	Q:37-94, H:33-86	44.8%	1.3 × 10^−6^	Conodipine-P3	36,110
DN3301 c0 g1 i1	KCP_HALAI	Q:132-479, H:5-120	49.1%	5.53 × 10^−41^	BPTI/Kunitz domain	10,069
DN126505 c0 g1 i1	CUE3_CONIM	Q:9-44, H:38-73	47.2%	1.38 × 10^−6^	Conotoxin Im14.3	29
DN13158 c3 g1 i1	COP1_CONPU	Q:27-93, H:28-90	44.8%	1.47 × 10^−12^	Conodipine-P1	1679
DN132248 c0 g1 i1	CCAP_CONVL	Q:46-124, H:22-96	50.6%	1.15 × 10^−17^	ConoCAP	28
DN133748 c0 g1 i1	CUE3_CONIM	Q:13-42, H:41-70	56.7%	5.69 × 10^−7^	Conotoxin Im14.3	42
DN17279 c3 g1 i1	CUE3_CONIM	Q:21-50, H:44-73	56.7%	8.29 × 10^−7^	Conotoxin Im14.3	27
DN109957 c0 g1 i1	CUE3_CONIM	Q:147-177, H:44-73	64.5%	6.39 × 10^−7^	Conotoxin Im14.3	641
DN18169 c0 g1 i3	CELE_CONVC	Q:21-125, H:1-98	60%	1.33 × 10^−27^	Elevenin-Vc1	100
DN18169 c0 g1 i2	CELE_CONVC	Q:21-125, H:1-98	60%	1.33 × 10^−27^	Elevenin-Vc1	91
DN19184 c2 g1 i2	COP1_CONPU	Q:7-183, H:8-176	31%	7.61 × 10^−26^	Conodipine-P1	68
DN19184 c2 g1 i1	COP1_CONPU	Q:7-183, H:8-176	31%	7.61 × 10^−26^	Conodipine-P1	19
DN4345 c0 g2 i2	CCAP_CONVL	Q:14-48, H:155-189	71.4%	1.11 × 10^−12^	ConoCAP	78
DN46921 c0 g1 i1	TU92_POLAB	Q:1-74, H:1-70	55.4%	2.78 × 10^−23^	Turripeptide Pal9.2	25
DN55000 c0 g1 i1	ACTP1_TERAN	Q:43-230, H:1-190	70%	3.68 × 10^−98^	Tereporin-Ca1	38
DN60625 c0 g1 i1	PV22_POMMA	Q:4-113, H:179-283	33.3%	6.28 × 10^−11^	Perivitellin-2 31 kDa sub.	21
DN87107 c0 g1 i1	ACTP1_TERAN	Q:1-62, H:127-190	65.6%	248 × 10^−11^	Tereporin-Ca1	34
DN94413 c0 g1 i1	CTHB5_CONVC	Q:55-149, H:1-96	80.2%	5.1 × 10^−51^	Thyrostimulin beta-5 sub.	22

* TPM: Raw values.

## Data Availability

Transcriptome data were deposited at the National Center for Biotechnology Information (bioproject: PRJNA1180544; biosample: SAMN44522766), and Mass spectrometry Appendix A can be downloaded at https://massive.ucsd.edu Data set: MSV000096243.

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
