# Peer review of "Molecular Insights into the Marine Gastropod Olivancillaria urceus: Transcriptomic and Proteopeptidomic Approaches Reveal Polypeptides with Putative Therapeutic Potential"

_ijms, 2025, doi:10.3390/ijms26083751_

Round 1

Reviewer 1 Report

Comments and Suggestions for Authors

This study employs a multi-omics approach to integrate transcriptomic and proteomic data from Olivancillaria urceus, investigating toxin transcript expression and its validation at the protein level. The research is biologically significant, with a large dataset and systematic bioinformatics analysis. However, there are inconsistencies in data presentation, insufficient methodological details, and issues with figure clarity that need to be addressed before publication.

  1. Inconsistencies in data presentation and figure references

1) Lines 149-152: “Subcategorization of the annotation based on GO terms provided by WEGO demonstrated consistent distribution with respect to cellular components, with the majority originating from the terms ‘Cell’ with 15,112 genes and ‘Organelle’ with 12,405 genes (Figure 3A).” The correct reference should be Figure 3B. 

2) Lines 154-156: “Additionally, sequences associated with ‘toxin activity’ were observed, guiding information of interest for bioprospecting with a total of 46 genes (Figure 3B).” The correct reference should be Figure 3C. 

3) Lines 161-163: “Finally, subcategorization in biological processes showed that most sequences are involved in cellular processes with 14,532 genes, metabolic processes with 11,061 genes, and biological regulation with 9,465 genes (Figure 3C).” The correct reference should be Figure 3A. 

4) Line 218: The first row in Table 1(line 170) records the Trinity ID as DN27199 c2 g1 i2, and line 218 also describes DN27199_c2_g1_i2. However, Figure 8 displays DN27199 c2 g1 i1. Please verify whether these IDs refer to the same gene or different transcripts. If they represent the same gene, ensure consistency; if they are different, provide clarification. 

5) Lines 399-402: The manuscript mentions DN27199 c2 g2 i1.p1, which is actually shown in Figure 8 (peptide fragments), but the text references Figure 7 (GO classification). Please verify whether this is a misreference and correct it in the text, or clarify why this transcript is mentioned in the context of GO classification. 

  1. Insufficient methodological details

1) Lines 489-491: The grouping of samples was based on body size rather than randomization, but the biological rationale for this choice is not explained. Please provide a justification for using body size as a grouping criterion instead of randomization. How does this approach affect the representativeness of the data? 

2) Lack of clarity in transcriptomic and proteomic data integration: The manuscript does not specify the criteria used to match transcriptomic and proteomic data. Were matches based on gene IDs, peptide sequences, or other parameters? Please describe the matching method in detail, including which bioinformatics tools were used and the specific parameters applied. 

3) Incomplete statistical details in mass spectrometry analysis: The manuscript states that an FDR threshold of ≤5% was used, but it does not provide comprehensive statistics on protein identification. Please include key protein identification statistics to enhance transparency and reproducibility. 

  1. Figure Improvements

1) Line 196: The legend does not clearly define the meanings of G1–G4. Please add an explanation of what G1–G4 represent (e.g., body size range) in either the figure legend or the main text. 

2) Line 241,278: Figure 8 and figure 10 have low resolution, making it difficult to interpret the data. Please provide a higher-resolution version of figures to improve clarity and readability. 

Author Response

Thank you very much for the constructive reviews and comments.

This study employs a multi-omics approach to integrate transcriptomic and proteomic data from Olivancillaria urceus, investigating toxin transcript expression and its validation at the protein level. The research is biologically significant, with a large dataset and systematic bioinformatics analysis. However, there are inconsistencies in data presentation, insufficient methodological details, and issues with figure clarity that need to be addressed before publication.

  1. Inconsistencies in data presentation and figure references

1) Lines 149-152: “Subcategorization of the annotation based on GO terms provided by WEGO demonstrated consistent distribution with respect to cellular components, with the majority originating from the terms ‘Cell’ with 15,112 genes and ‘Organelle’ with 12,405 genes (Figure 3A).” The correct reference should be Figure 3B. 

2) Lines 154-156: “Additionally, sequences associated with ‘toxin activity’ were observed, guiding information of interest for bioprospecting with a total of 46 genes (Figure 3B).” The correct reference should be Figure 3C. 

3) Lines 161-163: “Finally, subcategorization in biological processes showed that most sequences are involved in cellular processes with 14,532 genes, metabolic processes with 11,061 genes, and biological regulation with 9,465 genes (Figure 3C).” The correct reference should be Figure 3A. 

Response to questions 1, 2, and 3: We apologize for the mistake. The items in Figure 3 have been reorganized to match the order in which the information appears in the text. The figure legend has also been revised to reflect these changes. (Lines: 160-163)

4) Line 218: The first row in Table 1(line 170) records the Trinity ID as DN27199 c2 g1 i2, and line 218 also describes DN27199_c2_g1_i2. However, Figure 8 displays DN27199 c2 g1 i1. Please verify whether these IDs refer to the same gene or different transcripts. If they represent the same gene, ensure consistency; if they are different, provide clarification. 

Response: We appreciate your correction. The transcriptome, TPM, and proteome data were reviewed, and an error was found in the code. The correct code is DN27199 c2 g2 i1, and this has been updated in the text and tables. (Table 1 an 2; lines: 153,212,233,327,396)

5) Lines 399-402: The manuscript mentions DN27199 c2 g2 i1.p1, which is actually shown in Figure 8 (peptide fragments), but the text references Figure 7 (GO classification). Please verify whether this is a misreference and correct it in the text, or clarify why this transcript is mentioned in the context of GO classification. 

Response: Thank you very much for the correction. The reference in the text has been updated to Figure 8. (lines: 209, 397)

  1. Insufficient methodological details

1) Lines 489-491: The grouping of samples was based on body size rather than randomization, but the biological rationale for this choice is not explained. Please provide a justification for using body size as a grouping criterion instead of randomization. How does this approach affect the representativeness of the data?

Response: Since these were adult organisms collected directly from nature, it was difficult to determine their exact lifespan. The animals exhibited size differences, with a maximum variation of 1.5 cm in length and 1.1 cm in width. Groups G1, G2, and G3 had very similar sizes, while the animals in G4 were smaller. In this new version of the manuscript, we have added images of the animals used in the proteopeptidomic analysis (Supplementary Figure 1 and line: 192). Some differences were observed in the number of identified proteins between groups, but it is difficult to determine whether this was due to size differences, as variations were also present among the more similar groups.(lines: 485-488)

2) Lack of clarity in transcriptomic and proteomic data integration: The manuscript does not specify the criteria used to match transcriptomic and proteomic data. Were matches based on gene IDs, peptide sequences, or other parameters? Please describe the matching method in detail, including which bioinformatics tools were used and the specific parameters applied. 

Thank you very much for your comment. We have added more information to Section 4.4.4: Integrative Data Analyses of the Transcriptome and Proteome (lines 549-570)

3) Incomplete statistical details in mass spectrometry analysis: The manuscript states that an FDR threshold of ≤5% was used, but it does not provide comprehensive statistics on protein identification. Please include key protein identification statistics to enhance transparency and reproducibility. 

Response: Beyond the FDR threshold of ≤5, we also used additional criteria for key protein identification, including the Protein Score (-10lgP), which indicates the statistical significance of each protein (this value for each protein is shown in Supplementary Table 2). A higher value indicates greater confidence in the identification. Additionally, we considered the Mass Error Tolerance, which defines the tolerance in matching experimental and theoretical masses. The statistical results of proteomics and peptidomics generated by PEAKS Studio have now been added to the supplementary material 1 and referenced in the manuscript text. (line 188)

The FDR graphs for the proteome and peptidome reflect the quality of the data, as most of the identified peptides were classified with FDR ≤ 0.1% (72.6%), between 0.1% and 1.0% (14.4%), and between 1% and 5% (only 12.6%). The cutoff value used for the Protein Score was -10lgP ≥ 20, the Parent Mass Error Tolerance was 15.0 ppm, and the Fragment Mass Error Tolerance was 0.5 Da. When manually inspecting the sequencing data, we observed the presence of numerous b ions and y ions for peptides selected by the software, which were highlighted in blue in each gene sequence from the database, indicating high quality.

  1. Figure Improvements

1) Line 196: The legend does not clearly define the meanings of G1–G4. Please add an explanation of what G1–G4 represent (e.g., body size range) in either the figure legend or the main text.

Thank you very much. The information has been added to the figure legend. (lines:189-192)

2) Line 241,278: Figure 8 and figure 10 have low resolution, making it difficult to interpret the data. Please provide a higher-resolution version of figures to improve clarity and readability. 

Thank you very much for your observation regarding Figures 8 and 10. We have improved their resolution. (lines: 230, 267)

Reviewer 2 Report

Comments and Suggestions for Authors

This manuscript explores the transcriptomic and proteopeptidomic profiles of Olivancillaria urceus, a marine gastropod species often discarded as bycatch. The study aims to identify toxin-like sequences with potential therapeutic value using omics-based approaches. While the topic is interesting and the multi-omics strategy is relevant, the manuscript would benefit from a clearer research focus, stricter criteria for toxin-related annotations, and more cautious interpretation of the findings. Several parts of the discussion overstate functional claims that are not directly supported by the data. Additionally, some methodological aspects, such as sample design and transcriptome redundancy control, need further clarification to ensure reproducibility and scientific rigor.

Ln 35: This opening sentence is quite common in introductions, but it adds little scientific value. It would be better to start with something more specific or directly relevant to the research question.

Ln 41: Saying that bycatch has “no economic value” feels too absolute. Given recent advances in biodiversity and bioprospecting research, many bycatch species are being reassessed for their potential. Perhaps rephrasing it as “often lacking recognized economic value” would be more accurate.

Ln 106: It might be more appropriate to place the data availability information in the Data Availability section rather than within the Results.

Ln 112: The numbre of transcripts (319k) and predicted proteins (239k) seems unusually high for a non-model gastropod. This could be due to assembly redundancy, alternative splicing, or technical artifacts. It would be helpful if the authors explained this in the Discussion and clarified whether any redundancy-reduction steps (e.g., CD-HIT) were used.

Ln 118: A BUSCO completeness score below 60% suggests that a considerable portion of expected genes may be missing. While not unexpected for non-model species, it seems misleading to call this “good integrity.” A more balanced description would be appropriate.

Ln 160: Since the TPM values span one to two orders of magnitude, it’s important to indicate whether these are raw or log-transformed values. Otherwise, using them for clustering or comparisons might introduce bias due to scale differences.

Ln 276: The phrase “with descriptions like toxins” isn’t very precise. It would be clearer to specify whether these sequences actually include known toxin domains (e.g., Pfam), or are annotated as toxin-related entries in databases like UniProt or InterPro.

Ln 319: When discussing toxin-like domains, it’s important to clarify whether they were predicted based on sequence similarity or domain structure. Some domains like Kunitz are found in both toxins and non-toxin proteins, so a cautious interpretation is necessary.

Ln 324–355: This section reads more like a general summary of known marine toxins than a focused discussion. The main issue is that none of these functions are actually shown in the data from O. urceus—the claims are based solely on homology or predicted structure, which makes them speculative.

Ln 357–368: These functional descriptions (e.g., reducing heart rate or inducing hyperactivity) seem overly speculative without any experimental context or organism-specific data. They’re based on observations in other species and don’t appear directly supported here.

Ln 383, 387: Some of the claims in this part go well beyond what the presented data can justify. I’d recommend using a more neutral tone and framing these points as possibilities rather than conclusions.

Ln 430–444, It’s unclear whether the transcriptome sequencing was performed on pooled samples or a single sample. Since only one specimen is mentioned later, there may be concerns about biological variability. Clarifying the sampling strategy and whether replicates were included would strengthen the methods.

Ln 438: The manuscript should mention which tool or system was used to assess RNA quality (e.g., Bioanalyzer, TapeStation), since this affects data reliability downstream.

Comments on the Quality of English Language

The English throughout the manuscript is generally clear, but there are some awkward expressions and inconsistent use of terminology that may affect the overall readability. A light language revision by a fluent speaker could help improve clarity and flow.

Author Response

Thank you very much for the constructive reviews and comments. We appreciate your time and efforts.

This manuscript explores the transcriptomic and proteopeptidomic profiles of Olivancillaria urceus, a marine gastropod species often discarded as bycatch. The study aims to identify toxin-like sequences with potential therapeutic value using omics-based approaches. While the topic is interesting and the multi-omics strategy is relevant, the manuscript would benefit from a clearer research focus, stricter criteria for toxin-related annotations, and more cautious interpretation of the findings. Several parts of the discussion overstate functional claims that are not directly supported by the data. Additionally, some methodological aspects, such as sample design and transcriptome redundancy control, need further clarification to ensure reproducibility and scientific rigor.

Ln 35: This opening sentence is quite common in introductions, but it adds little scientific value. It would be better to start with something more specific or directly relevant to the research question.

Response: Thank you very much for the suggestion. The first sentence has been removed, and we now present the issue directly regarding bottom trawling and bycatch. 

Ln 41: Saying that bycatch has “no economic value” feels too absolute. Given recent advances in biodiversity and bioprospecting research, many bycatch species are being reassessed for their potential. Perhaps rephrasing it as “often lacking recognized economic value” would be more accurate.

Response: We agreed with the suggestion and revised the sentence accordingly. (line 39)

Ln 106: It might be more appropriate to place the data availability information in the Data Availability section rather than within the Results.

Response: Thank you. The sentence has been moved to the Data Availability section. (lines:598-599)

Ln 112: The number of transcripts (319k) and predicted proteins (239k) seems unusually high for a non-model gastropod. This could be due to assembly redundancy, alternative splicing, or technical artifacts. It would be helpful if the authors explained this in the Discussion and clarified whether any redundancy-reduction steps (e.g., CD-HIT) were used.

Response: Although the number of transcripts is high, it is within the same order of magnitude as other gastropod/mollusk transcriptomes reported in the literature. We have added some references in the discussion section to support this (lines 286-288). A large number of transcripts is common in transcriptomes assembled solely from short reads, and we agree with the points raised in your review: redundancy in the assembly, alternative splicing, and technical artifacts. However, no attempt was made to reduce redundancy, as the primary objective of the transcriptome assembly was to serve as a database for proteomic analysis. In this context, reducing redundancy in the transcriptome would lead to a lower number of identified peptides. A paragraph has been added to the Methods section to clarify this point. Additionally, beyond the toxins, most of the identified and discussed transcripts were consistent with the type of tissue analyzed. (lines: 430-432)

Ln 118: A BUSCO completeness score below 60% suggests that a considerable portion of expected genes may be missing. While not unexpected for non-model species, it seems misleading to call this “good integrity.”A more balanced description would be appropriate.

Response: We modified the description of BUSCO analysis parameters to achieve a more balanced assessment. (lines 114-123)

Ln 160: Since the TPM values span one to two orders of magnitude, it’s important to indicate whether these are raw or log-transformed values. Otherwise, using them for clustering or comparisons might introduce bias due to scale differences.

Response: Thank you very much for the comment. The TPM values presented are raw and have not been log-transformed. A note indicating this has been added to the footer of Table 1. These data were used in a purely descriptive manner, to illustrate the relative transcript abundance within this specific sample. The values were not used for clustering, comparisons, or other statistical analyses.

Ln 276: The phrase “with descriptions like toxins” isn’t very precise. It would be clearer to specify whether these sequences actually include known toxin domains (e.g., Pfam), or are annotated as toxin-related entries in databases like UniProt or InterPro.

Response: The requested information has been added. (lines:264-266)

Ln 319: When discussing toxin-like domains, it’s important to clarify whether they were predicted based on sequence similarity or domain structure. Some domains like Kunitz are found in both toxins and non-toxin proteins, so a cautious interpretation is necessary.

Response: Specifically for domains like Kunitz, it was mentioned in the discussion both the similarity of the domain as a protease inhibitor or as a toxin, highlighting the identity percentage found in the Blast described below: 

"A BLAST analysis showed 55.47%, 49.26%, and 47.79% identity with the TFPI-like multiple Kunitz type protease inhibitor sequence from the salivary gland of the non-conoidean neogastropod Colubraria reticulata (SPP68599.1; [44]) and with conkunitizins from Conus magus (DAC80551.1) and Conus ermineus (AXL95648.1), respectively."

Ln 324–355: This section reads more like a general summary of known marine toxins than a focused discussion. The main issue is that none of these functions are actually shown in the data from O. urceus—the claims are based solely on homology or predicted structure, which makes them speculative.

Ln 357–368: These functional descriptions (e.g., reducing heart rate or inducing hyperactivity) seem overly speculative without any experimental context or organism-specific data. They’re based on observations in other species and don’t appear directly supported here.

Ln 383, 387: Some of the claims in this part go well beyond what the presented data can justify. I’d recommend using a more neutral tone and framing these points as possibilities rather than conclusions.

Response: Thank you very much for the valuable comments. In response to the three points raised in the Discussion section, we have modified the title of this subsection to “Putative Toxin-Related Transcripts and Polypeptides in O. urceus”. Additionally, we have included the following paragraph: “Although these transcripts show sequence similarity to known toxins and contain conserved structural motifs associated with bioactive proteins, their functional roles in O. urceus have not yet been demonstrated. Future studies aiming to experimentally validate the activity, expression patterns, and potential ecological functions of these sequences will be essential to clarify their biological relevance in this species.” With this addition, we aim to emphasize that our discussion focuses on the sequence similarity between our transcripts and genes associated with toxin-like activity. We hope in this way maintains a neutral tone and presents these findings as potential associations rather than definitive conclusions. (lines:316;380-384)

Ln 430–444, It’s unclear whether the transcriptome sequencing was performed on pooled samples or a single sample. Since only one specimen is mentioned later, there may be concerns about biological variability. Clarifying the sampling strategy and whether replicates were included would strengthen the methods.

Response: Thank you very much for this comment. The transcriptome analysis was performed using a single sample derived from an adult animal. After consulting with our group's bioinformatician and considering the specific purpose of this transcriptome — to serve as a reference database for downstream proteopeptidomics experiments — we determined that a run with a higher read depth would have been more appropriate. Furthermore, due to the high cost of transcriptome sequencing, we opted instead to perform four proteomics and peptidomics runs, each using pooled samples from three animals. In the revised version of the manuscript, we have added the information regarding the use of a single sample for transcriptome sequencing in the Methods section, along with a note on the importance of achieving greater read depth. (lines 427; 430-432)

Ln 438: The manuscript should mention which tool or system was used to assess RNA quality (e.g., Bioanalyzer, TapeStation), since this affects data reliability downstream.

Response: Thank you very much. We have added the requested information. (lines 434-436) 

Round 2

Reviewer 1 Report

Comments and Suggestions for Authors

I think this current version has been significantly improved in revision. It can be accepted for publication in this journal.

Reviewer 2 Report

Comments and Suggestions for Authors

Thank you for addressing my review comments. The manuscript has been improved by your clarifications regarding methodology and the more measured interpretation of the toxin-like sequences. I appreciate your explanation about the high transcript number and sample design limitations.

While some concerns about the research design remain inherent to the approach taken, your revisions have helped to present the findings in a more appropriate scientific context.